# piR-hsa-022095 Drives Hypertrophic Scar Formation via KLF11-Dependent Fibroblast Proliferation

**DOI:** 10.3390/biomedicines13122963

**Published:** 2025-12-02

**Authors:** Rongxin Ren, Wenjiang Qian, Hongyi Zhao, Di Wang, Yanxia Xiao, Yajun Lin

**Affiliations:** 1Department of Plastic Surgery, Beijing Hospital, National Center of Gerontology, Institute of Geriatric Medicine, Chinese Academy of Medical Sciences, Beijing 100730, China; renrongxin4603@bjhmoh.cn (R.R.); qianwenjiang2927@bjhmoh.cn (W.Q.); zhaohongyi2642@bjhmoh.cn (H.Z.); 2The Key Laboratory of Geriatrics, Beijing Institute of Geriatrics, Institute of Geriatric Medicine, Chinese Academy of Medical Sciences, Beijing Hospital, National Center of Gerontology of National Health Commission, Beijing 100730, China; 18613316606@163.com (D.W.); xiaoyanxia2021@126.com (Y.X.)

**Keywords:** hypertrophic scar, fibroblast, proliferation, piR-hsa-022095, KLF11

## Abstract

**Background/Objectives:** Hypertrophic scar (HS) is a fibroproliferative disorder characterized by excessive fibroblast activation and collagen deposition. The role of PIWI-interacting RNAs (piRNAs) in HS pathogenesis has not been defined. This study aimed to identify HS-related piRNAs, clarify their molecular mechanisms, and evaluate their therapeutic potential. **Methods:** High-throughput piRNA sequencing was performed on hypertrophic scar and matched normal tissues, followed by validation in patient-derived samples and dermal fibroblasts using quantitative reverse transcription PCR. Functional assays, including proliferation, apoptosis, migration, and invasion assays, were conducted after transfection with piRNA mimics or inhibitors. RNA sequencing, Gene Ontology, and Kyoto Encyclopedia of Genes and Genomes enrichment analyses, as well as dual-luciferase reporter and rescue assays, were used to identify and confirm molecular targets. **Results:** Sequencing revealed piR-hsa-022095 as one of the most significantly upregulated piRNAs in HS. Its inhibition suppressed fibroblast viability, migration, and invasion while inducing apoptosis and G_0_/G_1_ arrest. Transcriptomic profiling identified cell-cycle–related genes as major downstream targets, with KLF11 emerging as the principal effector. piR-hsa-022095 targets the 3′ UTR of KLF11, repressing its expression and thereby facilitating fibroblast proliferation. Restoration of KLF11 reversed the pro-fibrotic effects of piR-hsa-022095, confirming its functional role in HS pathogenesis. **Conclusions:** This study identifies piR-hsa-022095 as a novel regulator implicated in HS formation through repression of KLF11. The piR-hsa-022095–KLF11 axis may represent a previously unrecognized regulatory pathway involved in hypertrophic scar formation, providing new insights into the molecular mechanisms underlying HS pathogenesis.

## 1. Introduction

Hypertrophic scars (HSs) are fibroproliferative lesions characterized by excessive fibroblast proliferation and migration, accompanied by abnormal extracellular matrix (ECM) accumulation [1,2]. They develop in up to 70% of patients following surgery, burns, or traumatic injury, causing functional and aesthetic impairment, psychological distress, and a marked reduction in quality of life. Current therapeutic approaches, including surgical excision, laser therapy, silicone gel application, pressure dressings, and corticosteroid injections, are largely nonspecific, yield limited long-term efficacy, and are prone to recurrence [3]. In recent years, multiple classes of non-coding RNAs—including microRNAs, long non-coding RNAs (lncRNAs), and circular RNAs (circRNAs)—have also been reported to be dysregulated in fibrotic skin conditions and to modulate fibroblast activation and ECM deposition [4]. Although dysregulation of transforming growth factor beta (TGF-β), phosphoinositide 3-kinase/protein kinase B (PI3K/Akt) signaling, and selected non-coding RNAs, and DNA methylation pathways has been implicated [5,6,7,8], the key molecular regulators of HS pathogenesis remain undefined, underscoring the need for targeted therapeutic strategies.

PIWI-interacting RNAs (piRNAs) represent a newly characterized group of small regulatory non-coding RNAs, typically 24–32 nucleotides in length. Initially discovered for their association with PIWI proteins in transposon silencing, they were first characterized for their essential role in germline development [9]. Beyond this canonical function, growing evidence implicates piRNAs as dynamic regulators in a wide range of human diseases, including cancer, cardiovascular disorders, neurodegenerative diseases, and viral infections [10,11,12,13]. Aberrant piRNA expression patterns have been reported in lung, gastric, breast, renal, and colorectal cancers, where they influence proliferation, migration, invasion, apoptosis, angiogenesis, and epithelial–mesenchymal transition [14]. Recent studies indicate that tissue-specific piRNA profiles modulate key signaling pathways at both transcriptional and post-transcriptional levels [15]. In the cytoplasm, piRNAs regulate gene expression by targeting complementary RNAs for translational repression or degradation, and recent studies suggest that such dysregulation may also contribute to aberrant tissue remodeling and fibrosis in non-cutaneous organs [11,16,17]. However, in contrast to microRNAs, lncRNAs, and circRNAs, the potential involvement of piRNAs in cutaneous fibrosis and hypertrophic scar formation has not yet been defined. Given their capacity for post-transcriptional and epigenetic regulation, piRNAs may represent a previously overlooked layer of ncRNA-mediated control in fibrotic remodeling.

Given the tumor-like fibroblast overgrowth observed in HS and the emerging roles of piRNAs in pathological tissue remodeling, we hypothesized that specific piRNAs may contribute to HS pathogenesis by modulating critical signaling pathways. To evaluate this assumption, we conducted high-throughput piRNA sequencing on HS tissue and matched normal skin, followed by validation in patient-derived samples and primary dermal fibroblasts. The objective of this study was to identify HS-associated piRNAs, define their molecular targets, and elucidate the signaling mechanisms by which they regulate fibroblast behavior, with the goal of advancing precision therapeutic strategies for HS.

## 2. Materials and Methods

### 2.1. Specimen Acquisition and Ethical Clearance

HS specimens and paired adjacent normal skin tissues (*n* = 20 patients) were obtained during reconstructive procedures in the Department of Plastic and Reconstructive Surgery, Beijing Hospital. Normal skin samples were collected from the same operative field, typically 0.5–1 cm beyond the visible scar margin and routinely excised during surgery. Both HS and normal skin tissues were histopathologically examined to verify diagnostic accuracy. All tissues were full-thickness (epidermis and dermis). Inclusion criteria: (1) Clinically and/or histopathologically confirmed hypertrophic scar tissue; (2) Scars resulting from surgical procedures; (3) Age ≥ 18 years; (4) Lesions located on the abdomen, back, or neck. Exclusion criteria: (1) Evidence of active or secondary infection at the operative site; (2) Prior anti-scar treatments that could affect tissue biology, including radiotherapy or pharmacologic interventions (e.g., systemic or intralesional corticosteroids). Each excised specimen was standardized to approximately 1 × 2 cm in size and divided into two portions: one for primary fibroblast isolation and the other for RNA extraction, which was immediately subjected to rapid freezing for storage.

All subjects gave documented consent after receiving a detailed explanation of the study’s aims and procedures. Approval for the research protocol was obtained from the Institutional Review Board of Beijing Hospital, with the assigned reference number 2024BJYYEC-KY042-02, and all procedures complied with the ethical standards of the 1964 Declaration of Helsinki and its later amendments.

### 2.2. Primary Skin Fibroblasts Isolation and Culture

Primary human dermal fibroblasts were isolated following a previously described enzymatic digestion protocol [18]. Briefly, fresh tissue specimens were minced into ~2 × 2 mm fragments and digested with type I collagenase (2 mg/mL; Sigma-Aldrich, Burlington, MA, USA) at 37 °C for 2 h. The pellet was resuspended in DMEM (Gibco, Waltham, MA, USA) containing 10% fetal bovine serum (FBS; Gibco). Cultures were maintained at 37 °C in a humidified 5% CO_2_ incubator, and only passages 3–5 were employed in experiments to preserve cellular phenotype.

### 2.3. Cell Transfection

Fibroblasts were exposed to either piRNA mimics or inhibitors (40 nM; Genomeditech, Shanghai, China) together with matched negative controls. Transfection was performed using Lipofectamine RNAiMAX reagent (Thermo Fisher Scientific, Waltham, MA, USA) following the protocol supplied by the manufacturer, with minor adjustments to optimize efficiency. The sequences used were as follows:

piR-hsa-022095 mimic: 5′-TACCGGAGCTGTGTGGTGTGTGGACGC-3′;

piR-hsa-022095 inhibitor: 5′-GCGUCCACACACCACACAGCUCCGGUA-3′.

In brief, cells were exposed to the transfection mixture for 6 h, then the existing culture medium was removed and substituted with fresh complete DMEM. Unless otherwise indicated, all experimental conditions were carried out in triplicate using independent biological samples. Cells were harvested at defined post-transfection intervals for downstream functional assays including cell viability, migration, and apoptosis, as well as molecular analyses, including quantitative PCR and Western blotting.

### 2.4. Cell Viability Analysis

Viability of transfected fibroblasts was determined using the Cell Counting Kit-8 (CCK-8; Sigma-Aldrich, Burlington, MA, USA), which is based on the WST-8 reagent. Cells (5 × 10^3^ per well, 96-well plates) received 10 µL reagent at 24, 48, and 72 h and were incubated for 4 h at 37 °C before spectrophotometric reading at 450 nm on an instrument (Bio-Rad, Hercules, CA, USA).

### 2.5. Cell Apoptosis and Cell Cycle Analysis

Apoptosis was assessed using Annexin V-FITC/propidium iodide (PI) staining (Beyotime, Shanghai China). Following treatment, fibroblasts were trypsinized with 0.25% trypsin, rinsed twice in chilled PBS, suspended and stained with Annexin V-FITC and PI for 15 min under light-protected conditions. The stained cell suspensions were subjected to flow cytometric detection using a BD LSRFortessa instrument (BD Biosciences, Milpitas, CA, USA), and apoptotic populations were quantified using FlowJo software (Version 10.0.7).

For cell-cycle profiling, treated fibroblasts were fixed and rinsed twice with cold PBS, then stained with PI/RNase solution for 30 min on ice. DNA content was measured on the BD LSRFortessa, and FlowJo was used to calculate the proportions of cells in G_0_/G_1_, S, and G_2_/M phases.

### 2.6. Quantitative Reverse Transcription Polymerase Chain Reaction (qRT-PCR)

Total RNA was isolated using TRIzol reagent (Invitrogen, Carlsbad, CA, USA) and quantified with a NanoDrop 1000 spectrophotometer (Thermo Fisher Scientific). For piRNA detection, cDNA was synthesized using a poly(A)-tailing and adapter-based reverse transcription method with a small-RNA–specific kit (Mir-X miRNA First-Strand Synthesis Kit, Takara, Shiga, Japan), following the manufacturer’s instructions. This strategy enables reliable reverse transcription of short piRNAs (24–32 nt). For mRNA, reverse transcription was performed with a conventional RT kit (Takara, Shiga, Japan). Quantitative PCR was performed with SYBR Premix Ex Taq (Takara) on a 7500 Fast DX Real-Time PCR system (Applied Biosystems, Foster City, CA, USA). Relative expression levels were determined using the 2^−ΔΔCt^ method, with U6 serving as the internal control for piRNA and GAPDH for mRNA. Primer sequences are given in Appendix A.

### 2.7. Western Blot

Proteins from treated fibroblasts were lysed in RIPA buffer supplemented with phenylmethylsulfonyl fluoride (PMSF; Beyotime). Protein yield was quantified via a BCA assay (Beyotime). Aliquots of 20 µg total protein per lane were resolved on 10% SDS–PAGE gels (Bio-Rad) and transferred to PVDF membranes (Millipore, Burlington, MA, USA). After blocking, membranes were probed with primary antibodies to c-MYC (Proteintech, 10828-1-AP, Wuhan, China) and GAPDH (Proteintech, 60004-1-IG). Signals were visualized by enhanced chemiluminescence, and band intensity was measured in ImageJ (1.52a, National Institutes of Health, Bethesda, MD, USA).

### 2.8. Double Luciferase Reporter Analysis

Luciferase reporter assays were conducted as previously described [19]. Reporter constructs for TSC22D2 and KLF11, each harboring the predicted piR-hsa-022095 binding sequence, were synthesized by GenePharma (Shanghai, China). Fibroblasts were co-transfected with the reporter plasmid, a Renilla luciferase control vector, and either the piR-hsa-022095 inhibitor or its matched negative control using Lipofectamine 3000 (Invitrogen). After 48 h, luminescence was recorded on an Affymetrix luminometer (Santa Clara, CA, USA) with the Dual-Luciferase Reporter Assay System (Promega, Madison, WI, USA). Firefly luciferase signals were normalized to Renilla activity.

### 2.9. Cell Motility Assays

The migratory and invasive capacities of fibroblasts were assessed with Transwell inserts equipped with 8-µm pore membranes (BD Biosciences). For invasion assays, 40 µL of Matrigel (BD Biosciences) was coated onto each insert, air-dried overnight, and rehydrated with Opti-MEM (Invitrogen) for 2 h. Migration assays were performed using uncoated inserts. Pre-treated fibroblasts (4 × 10^4^ cells) in serum-free Opti-MEM were placed into the upper chamber, while complete DMEM was placed in the lower compartment. For migration assays, the cells were incubated for 24 h at 37 °C in 5% CO_2_; for invasion assays, the incubation time was 48 h under the same conditions. Cells adhering to the underside of the insert were fixed, stained with 0.1% crystal violet (Beyotime) for 30 min, and imaged under an Olympus microscope (Tokyo, Japan). Migratory or invasive cells were counted in five randomly selected fields per insert.

### 2.10. Scratch Wound-Healing Assay

Fibroblasts were transfected with piR-hsa-022095 inhibitor or negative control as described above. When cell monolayers reached confluence, a linear scratch was created with a sterile 200-μL pipette tip, and detached cells were removed by PBS washing. Images were captured at 0 and 24 h under a phase-contrast microscope. The rate of wound closure was quantified using ImageJ software.

### 2.11. mRNA and piRNA Sequencing with Bioinformatic Analysis

Total RNA was extracted with TRIzol reagent (Invitrogen) from two types of experimental samples: (i) fibroblasts transfected with either the piR-hsa-022095 inhibitor or a negative control (*n* = 3 per condition) for transcriptome profiling, and (ii) three matched pairs of hypertrophic scar and adjacent normal skin for piRNA analysis.

For transcriptome and piRNA profiling, RNA was converted into protocol-specific cDNA libraries (mRNA libraries prepared according to standard transcriptome protocols and piRNA libraries synthesized and pre-amplified with Illumina-supplied primers) followed by sequencing on an Illumina NextSeq 500 platform (Genedenovo Biotechnology, Guangzhou, China).

Sequencing reads underwent Illumina quality control, adapter trimming, and filtering before alignment to the human reference genome for mRNA-seq or piRBase for piRNA-seq. Expression levels were normalized as transcripts per million (TPM) or counts per million (CPM). For mRNA analysis, differentially expressed genes (DEGs) were defined as those with |log_2_FC| ≥ 1 and an adjusted *p* < 0.05. Gene Ontology (GO) and Kyoto Encyclopedia of Genes and Genomes (KEGG) analyses were applied to annotate functions and identify enriched pathways of DEGs associated with piR-hsa-022095.

### 2.12. Statistical Analysis

Data are shown as mean ± standard deviation (SD) from ≥3 independent experiments. Statistical analyses were conducted in SPSS 20.0 and GraphPad Prism 6.0. Two-tailed paired or unpaired Student’s *t* tests were used for two-group comparisons, and one-way ANOVA with Tukey’s post hoc test for multiple groups. Spearman’s rank correlation assessed relationships between piR-hsa-022095 and KLF11 or TSC22D2 mRNA. *p* < 0.05 denoted significance.

## 3. Results

### 3.1. Dynamic Expression Landscape of piRNAs in HS Pathogenesis

To characterize piRNA alterations in hypertrophic scar (HS), we conducted small RNA sequencing on three HS lesions and their matched adjacent skin. High-quality reads were obtained from all six samples. Comparative analysis identified 178 differentially expressed (DE) piRNAs, including 107 upregulated and 71 downregulated species (Figure 1A). In line with earlier studies [20,21], most DE piRNAs measured approximately 30 nt in length (range 26–33 nt) (Figure 1B). Based on fold change (>1.5) and expression abundance, the top eight candidates were selected for validation in an independent cohort of 20 paired HS and control tissues. Quantitative RT-PCR confirmed differential expression for seven piRNAs (four increased, three decreased; Figure 1C–J), which were prioritized for subsequent functional studies.

### 3.2. Inhibition of piR-hsa-022095 Suppresses Fibroblast Proliferation and Motility While Inducing Apoptosis

To assess the functional relevance of the seven validated piRNAs, their mimics or inhibitors were introduced into primary dermal fibroblasts, and cell behavior was monitored using CCK-8 assays. Among all candidates, only piR-hsa-022095 inhibition caused a progressive, time-dependent reduction in cell viability (Figure 2A,B). Annexin V/PI staining further demonstrated significant increases in both early and late apoptotic fractions (Figure 2C,D). Flow cytometric analysis of PI-stained cells revealed that suppression of piR-hsa-022095 led to G_0_/G_1_ arrest accompanied by a marked decrease in the S-phase population (Figure 2E,F). In parallel, Transwell assays showed that piR-hsa-022095 knockdown markedly impaired fibroblast migratory and invasive capabilities (Figure 2G–J). To further assess fibroblast migration under wound-healing conditions, a scratch assay was performed. Consistent with the Transwell results, inhibition of piR-hsa-022095 significantly inhibited wound closure compared with the negative control (Figure 2K,L). Collectively, these findings indicate that piR-hsa-022095 promotes fibroblast growth and motility, while its inhibition triggering cell-cycle arrest and apoptosis.

### 3.3. piR-hsa-022095 Regulates a Network of Cell-Cycle–Associated Genes

To explore how piR-hsa-022095 exerts its downstream effects, we conducted RNA sequencing on fibroblasts transfected in the presence or absence of its inhibitor. This analysis identified 1787 differentially expressed mRNAs (740 up-regulated and 1047 down-regulated) (Figure 3A). GO-based functional profiling of downregulated transcripts revealed significant overrepresentation of pathways involved in cell proliferation, migration, adhesion, motility, and autophagy (Figure 3B). Intersection of these DEGs with predicted piR-hsa-022095 targets yielded 143 overlapping candidates (55 up-regulated, 88 down-regulated) (Figure 3C). KEGG analysis of this subset demonstrated enrichment for processes such as mitotic cell-cycle progression, cell-junction organization, and tissue morphogenesis (Figure 3D). qRT-PCR validation confirmed that piR-hsa-022095 inhibition upregulated four critical cell-cycle–related genes—PGPEP1, LMBR1L, TSC22D2, and KLF11 [22,23] (Figure 3E). Computational target prediction further revealed strong seed-sequence complementarity between piR-hsa-022095 and the transcripts of these genes (Figure 3F). These findings indicate that piR-hsa-022095 orchestrates a transcriptional program that promotes fibroblast proliferation and motility by repressing multiple cell-cycle–associated genes, with KLF11 emerging as a central downstream effector.

### 3.4. KLF11 Is the Molecular Target of piR-hsa-022095

To pinpoint the key downstream effector of piR-hsa-022095, we quantified the expression of the four candidate genes in ten paired HS and matched normal skin samples using RT-PCR. Both KLF11 and TSC22D2 were significantly reduced in HS lesions compared with controls (Figure 4A–D). However, only KLF11 levels exhibited a strong inverse correlation with piR-hsa-022095 abundance (Figure 4E,F). Dual-luciferase assays further confirmed that silencing piR-hsa-022095 markedly increased KLF11 reporter activity, whereas TSC22D2 activity was unaffected (Figure 4G). Together, these data identify KLF11 as the principal molecular target through which piR-hsa-022095 exerts its pro-fibrotic effects in HS.

### 3.5. KLF11 Restoration Reverses piR-hsa-022095–Induced Fibroblast Activation

KLF11 is known to repress SAMD7 and c-MYC within the TGF-β pathway, thereby inducing cell-cycle arrest or apoptosis [24]. We therefore investigated whether restoring KLF11 activity could counteract the pro-fibrotic effects of piR-hsa-022095 in HS. Among three siRNAs tested, si-KLF11#2 achieved approximately 50% knockdown efficiency, as determined by qRT-PCR (Figure 5A). Silencing KLF11 increased SAMD7 and c-MYC transcript levels and abolished their repression induced by piR-hsa-022095 inhibition (Figure 5B,C). Western blot analysis confirmed the restoration of c-MYC protein expression under these conditions (Figure 5D,E). Functionally, KLF11 depletion partially rescued fibroblast viability and cell-cycle progression suppressed by piR-hsa-022095 silencing (Figure 5F–H). Collectively, these findings position KLF11 as a putative downstream mediator of piR-hsa-022095–mediated fibroblast hyperproliferation in HS, highlighting a previously unrecognized regulatory pathway.

## 4. Discussion

Our study is the first to identify piR-hsa-022095 as a pro-proliferative small non-coding RNA that drives fibroblast activation in the pathogenesis of HS. We identify a novel mechanism in which upregulated piR-hsa-022095 suppresses KLF11-mediated TGF-β signaling, thereby promoting fibroblast hyperproliferation. Functionally, silencing piR-hsa-022095 reduced cell viability and motility while inducing apoptosis, whereas restoration of KLF11 reversed these effects. Together, these findings suggest that piR-hsa-022095–mediated fibroblast activation contributes to HS pathogenesis and identify the piR-hsa-022095–KLF11 axis as a potential molecular target warranting further investigation.

Dysregulated piRNA expression has been increasingly linked to diverse cancers [25,26], where they contribute to tumorigenesis and influence prognosis. Although HS is a non-neoplastic condition, it exhibits tumor-like behavior characterized by excessive dermal fibroblast proliferation. There is currently no published evidence addressing how piRNAs may contribute to hypertrophic scar pathophysiology. By applying high-throughput sequencing, we delineated dynamic piRNA expression profiles in HS tissue compared with matched normal skin. Of the eight candidate piRNAs identified, seven were validated in independent clinical samples. Among these, piR-hsa-022095 emerged as a functional driver of fibroblast proliferation, migration, and apoptosis.

A major focus of this work was uncovering the molecular target and downstream pathway of piR-hsa-022095. Consistent with the post-transcriptional function of piRNAs, which can interact with target mRNAs and inducing their degradation [27,28], RNA sequencing following piR-hsa-022095 inhibition revealed 1787 differentially expressed mRNAs, with cell-cycle genes significantly enriched by GO analysis. Subsequent validation identified KLF11 as a potential downstream target of piR-hsa-022095, whose regulation was confirmed by dual-luciferase reporter assays. Restoration of KLF11 expression rescued the phenotypic effects of piR-hsa-022095 inhibition, a finding in line with reports that Klf11-null mice develop marked fibrosis [29]. These results expand the mechanistic framework for piRNA-mediated gene regulation.

The TGF-β/Smad pathway is a well-established regulator of cell proliferation, differentiation, and apoptosis [30]. In HS, this cascade remains chronically activated, driving sustained fibroblast proliferation and collagen deposition that produce raised, rigid scar tissue [31,32]. KLF11, as a transcriptional repressor, modulates TGF-β activity partly through negative feedback involving Smad7 [33,34], thereby amplifying TGF-β signaling [35]. Our findings demonstrate that piR-hsa-022095 suppresses KLF11, thereby removing a key brake on TGF-β signaling and facilitating fibroblast proliferation. This previously unrecognized piR-hsa-022095–KLF11–TGF-β regulatory axis offers a mechanistic link between piRNA dysregulation and HS pathogenesis, and may serve as a promising molecular entry point for targeted scar therapies.

Despite the strengths of this study, several limitations should be acknowledged. The cohort size was modest and drawn from a single center, which may limit the broader applicability of our findings. Mechanistic conclusions were derived mainly from in vitro fibroblast assays; without in vivo validation, we cannot yet establish a causal role for piR-hsa-022095 in HS formation within the complex wound-healing microenvironment, where immune, vascular, and mechanical factors converge. This limitation should be taken into account when interpreting the pathogenic significance of our findings. Furthermore, the interaction between piR-hsa-022095 and KLF11 was inferred from dual-luciferase assays, which suggest post-transcriptional regulation but do not confirm direct binding. Further validation with mutant 3’UTR or PIWI pull-down assays is warranted. The upstream drivers of piR-hsa-022095 upregulation also warrant further investigation, with inflammation, mechanical stress, and epigenetic regulation as plausible contributors. Integrative multi-omics may further reveal upstream regulators and interacting non-coding RNAs, enabling a more complete framework for fibroblast activation in HS.

## 5. Conclusions

This study systematically profiled piRNA expression in HS tissue and identified piR-hsa-022095 as the first piRNA implicated in fibroblast proliferation in HS. piR-hsa-022095 promoted fibroblast growth, motility, and survival by directly repressing KLF11, thereby activating downstream SMAD7 and c-MYC signaling. Restoration of KLF11 expression reversed these effects, confirming it as a key mediator of piR-hsa-022095 activity. These findings suggest a previously unrecognized regulatory axis involved in HS pathogenesis and provide new mechanistic insights that may inform future diagnostic or therapeutic research.

## Figures and Tables

**Figure 1 biomedicines-13-02963-f001:**
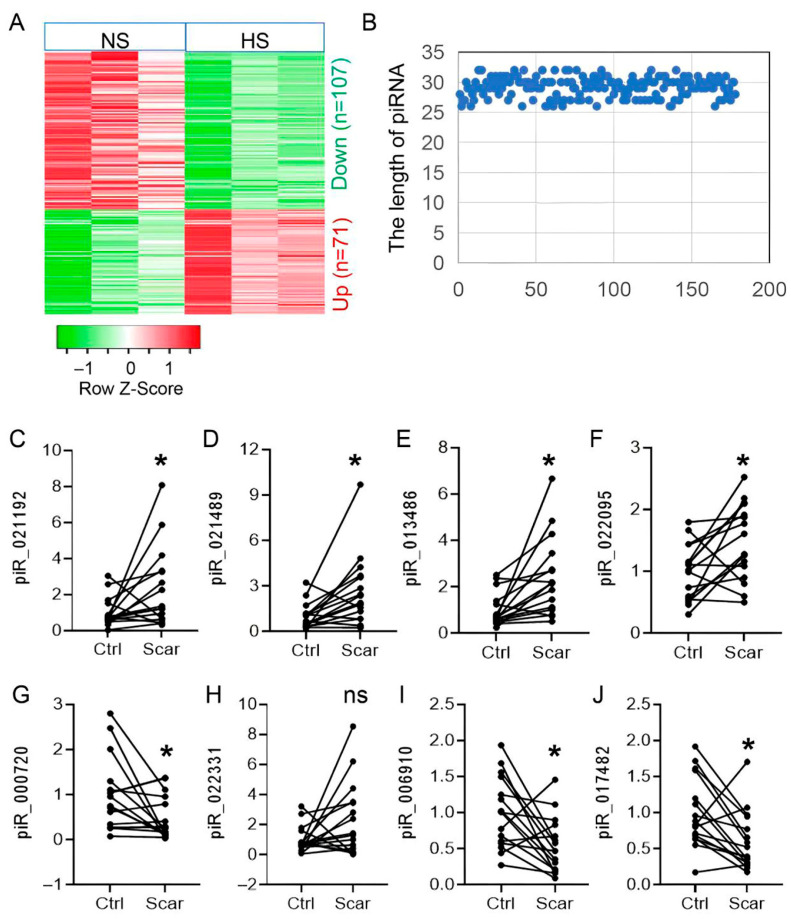
piR-hsa-022095 delineates the dynamic piRNA landscape in hypertrophic scar pathogenesis. (**A**) Heatmap of differentially expressed piRNAs (|log_2_FC| > 1, adjusted *p* < 0.05) in 3 paired hypertrophic scar (HS) and normal skin samples. (**B**) Length distribution of these piRNAs. (**C**–**J**) qRT-PCR validation of the top eight piRNAs in 20 paired HS/normal samples. Each dot represents one independent patient sample (mean of three technical replicates). * *p* < 0.05; ns, not significant (paired Student’s *t*-test).

**Figure 2 biomedicines-13-02963-f002:**
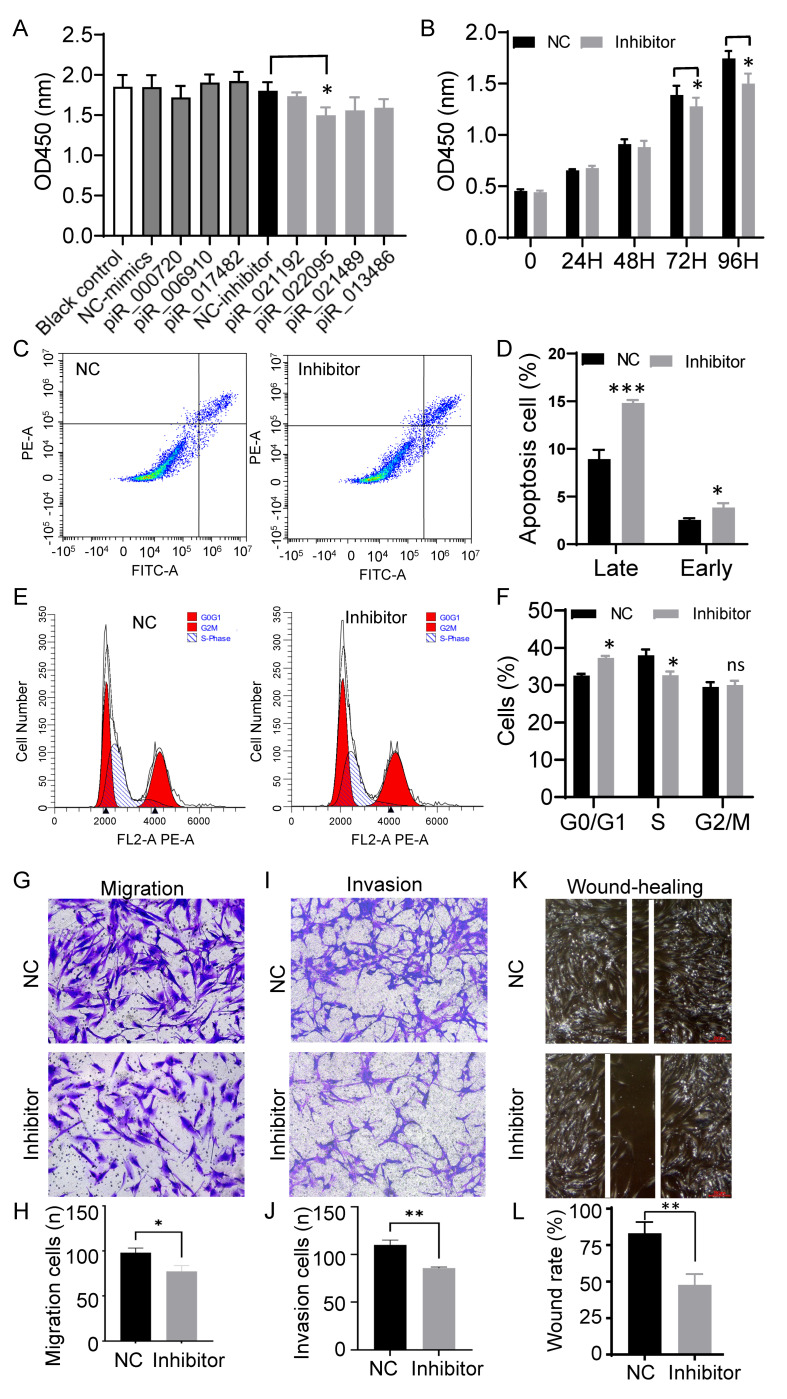
piR-hsa-022095 regulates fibroblast proliferation, apoptosis, and motility in vitro. (**A**) CCK-8 assay showing cell viability after transfection with piRNA mimics or inhibitors. (**B**) Time-course CCK-8 assay following piR-hsa-022095 knockdown. (**C**,**D**) Annexin V/PI flow cytometry demonstrating increased apoptosis upon piR-hsa-022095 inhibition. (**E**,**F**) PI-based cell-cycle analysis showing G_0_/G_1_ arrest and reduced S-phase. (**G**–**J**) Transwell assays revealing decreased migration and invasion after piR-hsa-022095 inhibition. (**K**,**L**) Scratch wound-healing assay of fibroblasts after piR-hsa-022095 inhibition. Data are presented as mean ± SD. * *p* < 0.05; ** *p* < 0.01; *** *p* < 0.001; ns, not significant (unpaired Student’s *t*-test).

**Figure 3 biomedicines-13-02963-f003:**
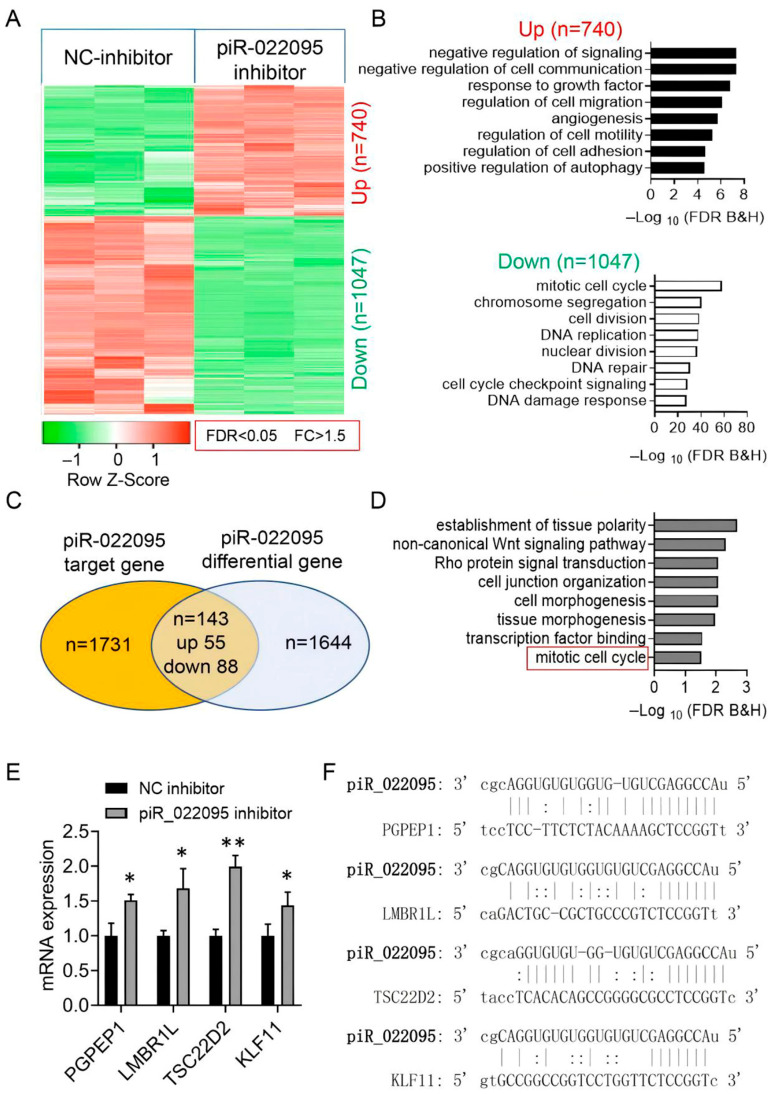
piR-hsa-022095 orchestrates a cell-cycle gene network. (**A**) Heatmap showing differentially expressed mRNAs after piR-hsa-022095 inhibition. (**B**) GO enrichment analyses of up- and down-regulated genes. (**C**) Overlap (Venn diagram) between regulated genes and predicted targets. (**D**) KEGG pathway enrichment of overlapping genes. (**E**) qRT-PCR confirming up-regulation of PGPEP1, LMBR1L, TSC22D2, and KLF11. Bars represent the mean ± SD of three independent biological replicates, each measured in technical triplicate. (**F**) Seed-sequence complementarity between piR-hsa-022095 and these genes. Data are presented as mean ± SD. * *p* < 0.05; ** *p* < 0.01 (unpaired Student’s *t*-test).

**Figure 4 biomedicines-13-02963-f004:**
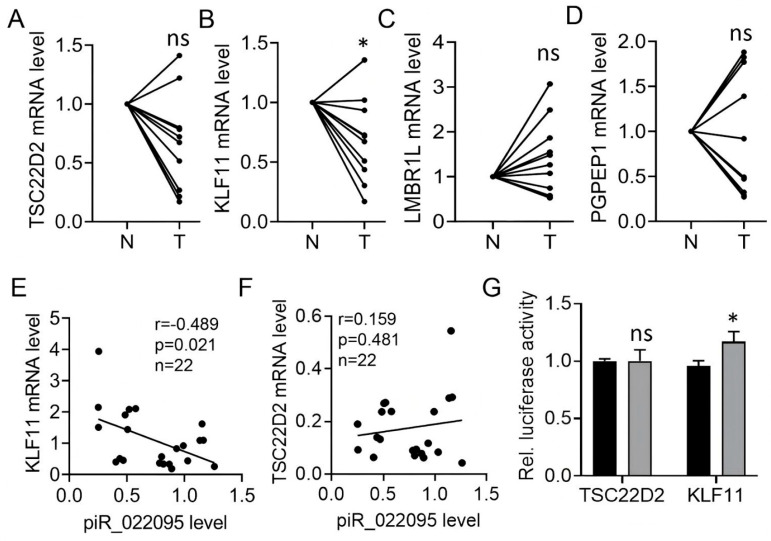
KLF11 is the molecular target of piR-hsa-022095. (**A**–**D**) qRT-PCR analysis of four cell-cycle–related genes in 10 paired HS and normal skin samples. (**E**,**F**) Spearman correlation between piR-hsa-022095 expression and KLF11 (**E**) or TSC22D2 (**F**) mRNA levels. (**G**) Dual-luciferase reporter assays showing increased KLF11 reporter activity but no significant effect on TSC22D2 after piR-hsa-022095 inhibition. Data are presented as mean ± SD. * *p* < 0.05; ns, not significant. (Statistics: paired Student’s *t*-test for (**A**–**D**); unpaired Student’s *t*-test for (**G**)).

**Figure 5 biomedicines-13-02963-f005:**
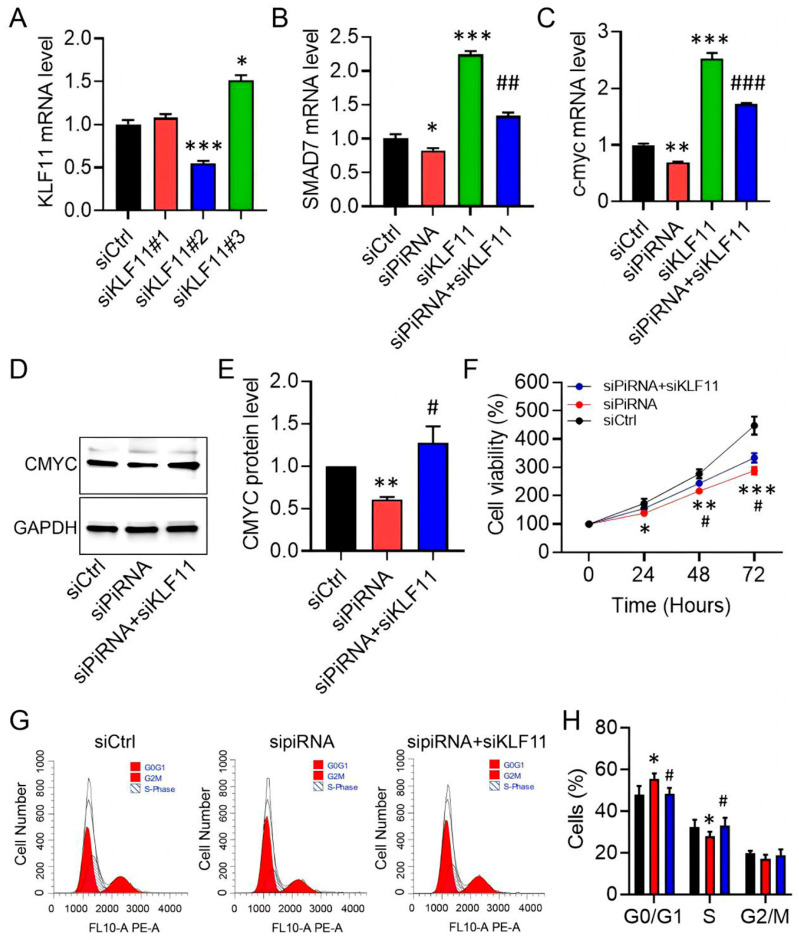
KLF11 restoration rescues piR-hsa-022095-driven phenotypes. (**A**–**C**) qRT-PCR of KLF11, SAMD7, and c-MYC after piR-hsa-022095 and/or KLF11 knockdown. (**D**,**E**) Representative immunoblot and quantification of c-MYC protein in fibroblasts. (**F**) CCK-8 viability under the indicated knockdowns. (**G**,**H**) Flow-cytometric DNA content analysis of cell-cycle progression. Data are presented as mean ± SD. * *p* < 0.05; ** *p* < 0.01; *** *p* < 0.001 vs. siCtrl; # *p* < 0.05; ## *p* < 0.01; ### *p* < 0.001 vs. siKLF11 (one-way ANOVA with Bonferroni post hoc test).

## Data Availability

Data supporting the findings of this study will be provided by the corresponding author upon justified request.

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
