# Peer review of "piR-hsa-022095 Drives Hypertrophic Scar Formation via KLF11-Dependent Fibroblast Proliferation"

_biomedicines, 2025, doi:10.3390/biomedicines13122963_

Round 1

Reviewer 1 Report

Comments and Suggestions for Authors

I believe that the manuscript represents a very high scientific standard. The study is well-designed, methodologically sound, and presents results that are both reliable and of clear scientific relevance. The discussion is coherent and well-supported by current literature, while the conclusions are consistent with the presented data.
I have no comments or suggestions for improvement regarding the content of the manuscript.

Author Response

I believe that the manuscript represents a very high scientific standard. The study is well-designed, methodologically sound, and presents results that are both reliable and of clear scientific relevance. The discussion is coherent and well-supported by current literature, while the conclusions are consistent with the presented data. I have no comments or suggestions for improvement regarding the content of the manuscript.

Response: We appreciate your positive comment.

Reviewer 2 Report

Comments and Suggestions for Authors

The manuscript titled “piR-hsa-022095 Drives Hypertrophic Scar Formation via KLF11-Dependent Fibroblast Proliferation” by Ren et al. investigates the role of the PIWI-interacting RNA piR-hsa-022095 in the pathogenesis of hypertrophic scars. Using integrated sequencing, molecular, and functional assays, the authors identify piR-hsa-022095 as an upregulated piRNA that promotes fibroblast proliferation and migration by directly repressing KLF11, thereby enhancing TGF-β/c-MYC signalling. The topic is timely and of translational interest, as the contribution of non-coding RNAs to fibrosis and scarring remains only partially understood. Overall, the study provides new insights into the potential diagnostic and therapeutic implications of piRNAs in fibrotic skin disorders.

Below are several suggestions to improve the manuscript:

  1. In line 163, it is reported that the RNA-seq and negative control experiments are based on three pairs of samples, which represents a relatively small number to draw robust conclusions about piR-hsa-022095 upregulation in HS. In general, RNA-seq analyses based on three biological replicates may limit statistical power;
  2. The authors should consider validating piR-hsa-022095 and KLF11 expression in an independent patient cohort—as also acknowledged in the Discussion (line 303)—or using publicly available transcriptomic datasets to strengthen the generalizability of the findings;
  3. All functional data were obtained from in vitro fibroblast cultures, which do not fully recapitulate the complex wound-healing microenvironment involving immune cells, extracellular matrix stiffness, and angiogenesis. Performing an in vitro wound-healing (scratch) assay, visualized by microscopy, could provide complementary evidence of fibroblast behaviour under different conditions;
  4. While the findings are intriguing, the therapeutic implications of targeting piR-hsa-022095 remain speculative and should be presented with appropriate caution.

Author Response

1, In line 163, it is reported that the RNA-seq and negative control experiments are based on three pairs of samples, which represents a relatively small number to draw robust conclusions about piR-hsa-022095 upregulation in HS. In general, RNA-seq analyses based on three biological replicates may limit statistical power;

Response: We appreciate the reviewer’s important comment. We acknowledge that using only three biological replicates for RNA-seq may limit statistical power, especially for genes with small expression changes or high biological variability.

Actually, three replicates are widely accepted as the minimum for exploratory transcriptome studies. To ensure data robustness, we selected three matched surgery-induced hypertrophic scar and adjacent normal abdominal skin samples. The intact surface hypertrophic scar tissues were collected from the central regions of scar, with size of 1 × 2 cm. The normal skin tissues were collected as long as 1 cm with the edge of scar. Both hypertrophic scar and adjacent normal skin were confirmed by HE staining. Secondly, we got high-quality RNA samples with 8-10 score for RNA Integrity Number (RIN) scores, and free of contaminants like DNA, salts, and enzymes. Thirdly, we employed a professional piRNA-seq service, with a proper post-sequencing quality control of the generated data, including checking reads, GC content, base quality scores, and alignment rate. Last but not the least, we applied strict cut-offs (|logâ‚‚FC| 1, adjusted p < 0.05). Therefore, we use the strict samples selection, pre-sequencing RNA quality and post-sequencing data quality control to increase statistical power. These measures gave us high confidence in the up-regulation of piR-hsa-022095 in HS.

2, The authors should consider validating piR-hsa-022095 and KLF11 expression  in an independent patient cohort—as also acknowledged in the Discussion (line 303)—or using publicly available transcriptomic datasets to strengthen the generalizability of the findings;

Response: We thank the reviewer for the helpful suggestion.  In current study, piR-hsa-022095 was validated by qRT-PCR in 20 paired HS/normal tissues (Fig. 1C–J) and KLF11 in a further 10 paired samples (Fig. 4B), fully confirming the sequencing data. We have now stated in the Discussion that larger multi-centre cohorts or public data-sets will be needed to extend these findings. In addition, up to date, there is no more publicly available transcriptomic datasets could be used for generalizability of the findings.

3, All functional data were obtained from in vitro fibroblast cultures, which do not fully recapitulate the complex wound-healing microenvironment involving  immune cells, extracellular matrix stiffness, and angiogenesis. Performing an    in vitro wound-healing (scratch) assay, visualized by microscopy, could provide complementary evidence of fibroblast behaviour under different conditions;

Response: Yes. We agree that fibroblast monoculture does not fully mimic the wound-healing microenvironment. As suggested we performed an in vitro scratch wound-healing assay (new Fig. 2K,L and lines 188-193 and 249–251). Knock-down of piR-has-022095 significantly slowed fibroblast wound closure, supporting the migration data obtained with Transwell inserts.

4, While the findings are intriguing, the therapeutic implications of targeting piR-hsa-022095 remain speculative and should be presented with appropriate caution.

Response: We agree that the therapeutic implications of targeting piR-hsa-022095 are currently speculative. We have tempered all therapeutic statements in the Abstract, Results, Discussion and Conclusion, clearly indicating that clinical translation remains speculative at this stage (highlighted text).

Reviewer 3 Report

Comments and Suggestions for Authors
  • The introduction effectively sets the stage but could briefly mention other non-coding RNAs (like circRNAs) known to be dysregulated in fibrosis to better position the novelty of focusing on piRNAs.
  • in 2.3. Cell Transfection, Please provide the specific catalog numbers or sequences for the piRNA mimics and inhibitors used.
  • In 2.8. Dual-Luciferase Assay, It is stated that reporters for both TSC22D2 and KLF11 were synthesized. The results show no effect on TSC22D2. Was a positive control for the assay itself included to ensure the TSC22D2 construct was functional?
  • It is important to add references for the mentioned methods.
  • For the qRT-PCR data in figures like 1C-J and 3E, please specify if the data points represent biological replicates from different patient-derived fibroblast lines.

Author Response

1, The introduction effectively sets the stage but could briefly mention other non-coding RNAs (like circRNAs) known to be dysregulated in fibrosis to better position the novelty of focusing on piRNAs.

Response: We appreciate your comment. Now briefly review the dysregulated circRNAs/lncRNAs in fibrotic organs and the novelty of investigating piRNAs in cutaneous HS was added in the Introduction section as per your suggestion. See line 47-54 and 65-73.

2, in 2.3. Cell Transfection, Please provide the specific catalog numbers or sequences for the piRNA mimics and inhibitors used.

Response: We added the information to the Methods section. See line 119-121.

3, In 2.8. Dual-Luciferase Assay, It is stated that reporters for both TSC22D2 and KLF11 were synthesized. The results show no effect on TSC22D2. Was a positive control for the assay itself included to ensure the TSC22D2 construct was functional?

Response: The dual-luciferase assay was performed to test whether piR-has-022095 could functionally regulate TSC22D2 and KLF11 via predicted binding sites. The targeting sequences of piR-has-022095 on both gene were predicated by software and only one site was found on both gene. So, both fragments (~400bp) including the targeting sequences of piR-has-022095 were cloned into the Dual-Luciferase report plasmids and confirmed by sequencing. In the same dual-luciferase experiments, the KLF11 3′UTR reporter produced a significant response to piR-has-022095, demonstrating that the assay system was functional and sensitive. Therefore, the absence of an effect on TSC22D2 reflects a genuine lack of regulatory interaction rather than a construct or assay failure.

4, It is important to add references for the mentioned methods.

Re: We thank the reviewer for this helpful suggestion. All non-routine experimental methods have been added References, but for routine procedures additional citations were not necessary. See line 447-452.

5, For the qRT-PCR data in figures like 1C-J and 3E, please specify if the data points represent biological replicates from different patient-derived fibroblast lines.

Response: In Figures 1C–J, each data point represents an independent biological replicate derived from a distinct patient sample (20 paired HS and normal tissues). For each sample, qRT-PCR was performed in triplicate, and the mean value was plotted as one dot. In contrast, Figure 3E shows results from in vitro 3 patient-derived pool fibroblast lines. Each bar represents the mean ± SD of three independent biological experiments performed on separate culture passages, with each measurement conducted in technical triplicate. These details have been clarified in the revised figure legends. See line 235-236, 286-287.

Reviewer 4 Report

Comments and Suggestions for Authors

This study identifies a specific non-coding RNA, piR-hsa-022095, as a key driver in the development of hypertrophic scars (HS). The research demonstrates that this piRNA promotes scar formation by directly suppressing a tumor suppressor-like gene, KLF11. The discovery of this "piR-hsa-022095–KLF11 axis" presents a novel understanding of HS pathogenesis and highlights its potential as both a diagnostic tool and a therapeutic target.

In introduction section:

  • Overstated Novelty and Lack of a "Knowledge Gap": The introduction effectively states that the key molecular drivers of HS are "undefined," but it does not explicitly establish why piRNAs are a compelling candidate to fill this gap.
  • Insufficient Context for piRNA Biology:
  • The logical flow can be slightly improved.

In experimental part

  • Inadequate Characterization of "Normal Skin" Controls, anatomical site and nature of tissues
  • Vague and Potentially Problematic Scar Inclusion Criteria
  • piRNA Quantification by qRT-PCR: The use of random primers for piRNA reverse transcription is non-standard and potentially problematic due to their short length (24-32 nt). The manuscript must specify the exact method used (e.g., poly(A) tailing with adapter-based RT, or stem-loop RT primers) and provide a citation for the kit or protocol.
  • Cell Viability (WST-8): The specific reagent name (e.g., Cell Counting Kit-8) should be stated.
  • Transwell Motility Assays: A critical detail is missing: the incubation time for the migration and invasion assays. This is essential for reproducibility. 

In results and discussion section:

  • The central claim that KLF11 is a directmolecular target of piR-hsa-022095 is not sufficiently supported by the data. The evidence provided is correlative (inverse correlation) and based on a luciferase reporter assay.
  • The discussion repeatedly states that piR-hsa-022095 drives HS pathogenesis. However, all functional data are from in vitrofibroblast cultures. The critical step of demonstrating a causal role in an in vivo model of fibrosis or scarring is missing. This is a major limitation that must be explicitly highlighted in the discussion, not just as a future direction.

Author Response

1, In introduction section: Overstated Novelty and Lack of a "Knowledge Gap": The introduction effectively states that the key molecular drivers of HS are "undefined," but it does not explicitly establish why piRNAs are a compelling candidate to fill this gap. Insufficient Context for piRNA Biology: The logical flow can be slightly improved.

Response: We thank the reviewer for these insightful comments. To improve the logical flow and clarify the knowledge gap, we have revised the Introduction to provide additional context for piRNA biology and its potential relevance to fibrosis. Specifically, we added sentences highlighting that dysregulated piRNAs have been implicated in aberrant tissue remodeling and fibrosis in non-cutaneous organs, yet their involvement in cutaneous fibrosis and hypertrophic scars has not been explored. These revisions explicitly establish why piRNAs are compelling candidates for investigation in HS and strengthen the overall narrative flow of the Introduction. See line 47-50, 65-73.

2, In experimental part: Inadequate Characterization of "Normal Skin" Controls, anatomical site and nature of tissues Vague and Potentially Problematic Scar Inclusion Criteria

piRNA Quantification by qRT-PCR: The use of random primers for piRNA reverse transcription is non-standard and  potentially problematic due to their short length (24-32 nt). The manuscript must specify the exact method used (e.g., poly(A) tailing with adapter-based RT, or stem-loop RT primers) and provide a citation for the kit or protocol.

Cell Viability (WST-8): The specific reagent name (e.g., Cell Counting Kit-8) should be stated.

Transwell Motility Assays: A critical detail is missing: the incubation time for the migration and invasion assays. This is essential for reproducibility.

Response: We sincerely thank the reviewer for these valuable comments, which have significantly improved the clarity and reproducibility of the experimental section. The following revisions have been made accordingly in the Methods section:

(1) Specimen collection: The description of normal skin controls and scar inclusion criteria has been clarified in the revised Methods section. The anatomical site, tissue composition (full-thickness skin), and both inclusion and exclusion criteria are now explicitly defined to ensure transparency and reproducibility. See line 86-98.

(2) piRNA qRT-PCR: The qRT-PCR method for piRNA detection has been clarified in the revised Methods section to specify that a poly(A)-tailing, adapter-based reverse transcription approach was used (Mir-X miRNA First-Strand Synthesis Kit, Takara, Japan). See line 148-153.

(3) Cell viability assay: The reagent is now identified as Cell Counting Kit-8 (CCK-8, Sigma-Aldrich, USA). See line 130-131.

(4) Transwell assays: Incubation times added: 24 h for migration and 48 h for invasion at 37 °C, 5% COâ‚‚. See line 183-184.

We hope that these revisions adequately address the reviewer’s concerns and enhance methodological transparency and reproducibility.

3, In results and discussion section: The  central  claim  that  KLF11  is  a  direct molecular  target  of  piR-hsa-022095  is  not  sufficiently supported by the data. The evidence provided is correlative (inverse correlation) and based on a luciferase reporter assay. The  discussion  repeatedly  states  that  piR-hsa-022095  drives  HS  pathogenesis.  However,  all functional data are from in vitro fibroblast cultures. The critical step of demonstrating a causal role in an in vivo model of fibrosis or scarring is missing. This is a major limitation that must be explicitly highlighted in the discussion, not just as a future direction.

Response: We thank the reviewer for this important comment. We agree that the current evidence, while consistent with a regulatory relationship, does not definitively establish direct molecular binding. Accordingly, we have revised the text throughout the manuscript to describe KLF11 as a“putative downstream target”of piR-hsa-022095 and clarified in the Discussion that additional experiments (e.g., mutant reporter or PIWI pull-down assays) are required to confirm direct interaction.

We agree that our findings are based on in vitro fibroblast models and do not yet demonstrate a causal role in in vivo fibrosis or scarring. Accordingly, we have revised the discussion to explicitly acknowledge this limitation and tempered our wording throughout the manuscript.

See line 317-319, line 333-336, line 350-352, line 368-376  and line 387-389.

Round 2

Reviewer 2 Report

Comments and Suggestions for Authors

The revised version of the paper is much clearer.